# Structural and phylogenetic insights from complete chloroplast genomes of seven *Vicia* species

**Mohammad Mehdi Golchini, Aboozar Soorni**\*

Department of Biotechnology, College of Agriculture, Isfahan University of Technology, Isfahan, Iran

\* soorni@iut.ac.ir

## Abstract

The legume genus *Vicia* L. (Fabaceae) is of significant ecological and agronomic importance, comprising species widely utilized as forage crops, green manure, and sources of valuable phytochemicals. Despite this, a comprehensive genomic understanding of many species, particularly those endemic to underrepresented regions like Iran, remains limited. To address this, we employed a high-throughput sequencing and comparative genomics approach to elucidate the chloroplast (cp) genome architecture and evolutionary relationships of seven previously uncharacterized Iranian *Vicia* species including *V. hirsuta*, *V. hybrida*, *V. lathyroides*, V. lutea, *V. narbonensis, V. peregrina*, and *V. villosa*. Total genomic DNA was sequenced on an Illumina HiSeq 2000 platform, and the cp genomes were assembled de novo using GetOrganelle, followed by comprehensive annotation with a suite of bioinformatic tools. The analysis revealed considerable size variation, ranging from 118,660–130,223 bp, and a key structural divergence involving the loss of one inverted repeat (IR) region in six species, consolidating their placement within the IR-lacking clade (IRLC), while *V. villosa* retained the ancestral quadripartite structure. Lineage-specific gene losses were documented, including *accD* in *V. lathyroides* and *ycf2* in *V. narbonensis*. Microsatellite analysis identified a predominance of A/T-rich mononucleotide simple sequence repeats (SSRs), with *V. hybrida* exhibiting the highest SSR density. Nucleotide diversity (Pi) analysis across coding regions identified *clpP* (Pi = 0.19772) and *ycf1* (Pi = 0.16964) as hypervariable loci, while the ribosomal protein genes *rps7* and *rpl20* were validated as highly effective phylogenetic barcodes. Maximum likelihood phylogenetic reconstruction, based on a concatenated alignment of 86 shared protein-coding genes, resolved the species into well-supported clades, providing a robust evolutionary framework. This study delivers essential genomic resources that deepen the understanding of cp genome evolution in the IRLC and provides powerful molecular tools for future research in *Vicia* systematics, conservation genetics, and precision breeding.

**Data availability statement:** The assembled and annotated genomes are accessible in NCBI database under the research accessions PV364429, and PV480546-PV480551.

**Funding:** The author(s) received no specific funding for this work.

**Competing interests:** The authors have declared that no competing interests exist.

## Introduction

The genus *Vicia* L., a member of the Fabaceae (Leguminosae) family, represents the third-largest group of flowering plants globally. Comprising approximately 150–210 species, it is widely distributed across Europe, Asia, and North America, with the highest species concentration found in the Mediterranean region [1,2]. Indeed, the Mediterranean is recognized as the primary center of diversification for *Vicia*, with Turkey and northwest Asia exhibiting the greatest species diversity [3,4]. *Vicia* species are predominantly cultivated as winter forage legumes due to their high nutritional value, serving as green manure, pasture, silage, and hay. Their adaptability to diverse climatic and soil conditions makes them particularly suitable for intercropping with cereals, where they contribute to disease suppression and soil improvement [5]. Additionally, these plants exhibit shade tolerance and possess nitrogen-fixing capabilities, enriching the soil with bioavailable nitrogen that benefits neighboring crops [6]. Beyond their agricultural significance, many Fabaceae species hold considerable economic value as sources of food, herbal medicine, industrial materials, and animal feed [7]. Notably, Vicia seed protein concentrate has recently emerged as a viable raw material for producing vacuum thermoformed bioplastics, demonstrating acceptable mechanical resistance and stability [8]. Furthermore, *Vicia* seeds contain a diverse array of bioactive compounds, including phenolic acids, flavonoids, organic acids, hydroxybenzoic aldehydes, amino acids, lignans, and terpenes. These phytochemicals underscore their potential applications in pharmaceuticals and functional food additives [9–14]. Scientific investigations have confirmed that *Vicia* seeds exhibit significant antioxidant properties, nitric oxide scavenging activity, metal chelation capacity, and enzyme inhibitory effects, further supporting their therapeutic potential [6].

Given the ecological and economic significance of *Vicia* species, a deeper understanding of their genetic diversity and evolutionary relationships is essential for optimizing their agricultural and biotechnological applications. Chloroplasts, as central organelles in plant metabolism, play a pivotal role in this context. They are indispensable for plant survival, converting solar energy into chemical energy through photosynthesis, while their genomes encode critical genes for photosynthesis and other metabolic processes [15–18]. Advances in cp genomics have profoundly enriched plant biology, offering insights into species diversity, evolutionary dynamics, and functional adaptation, knowledge that is instrumental in crop improvement and biotechnology. Structurally, the cp genome of land plants is characterized by a conserved quadripartite organization, comprising a large single-copy (LSC) region, a small single-copy (SSC) region, and two inverted repeat (IR) regions. Although typically circular and ranging between 107 kb and 218 kb in size, these genomes exhibit notable structural variability, including IR loss or gene family deletions, despite their overall stability. They commonly harbor 120–130 genes, predominantly associated with photosynthesis, transcription, and translation, reflecting their functional specialization. The utility of cp genome sequences extends across multiple disciplines. In phylogenetics, they serve as robust markers for resolving evolutionary relationships among species, while domestication studies leverage them to trace the origins and genetic

shifts in cultivated plants. Chloroplast-derived DNA barcodes are particularly valuable for cultivar identification and the conservation of genetic resources, aiding in the protection of agrobiodiversity. Beyond traditional breeding, cp genomes are increasingly exploited in biotechnology, where their engineered variants enable the development of stress-resistant crops and the production of recombinant proteins, including vaccines and biopharmaceuticals, in edible plant systems [19–21].

Comparative genomic analyses of *Vicia* species demonstrate a consistent departure from the typical quadripartite chloroplast structure, primarily due to the loss of IRs. This structural simplification results in a tripartite genome organization, exemplified by *V. bungei* (130,796 bp), *V. sepium* (124,095 bp), and *V. faba* (122,569 bp) [22,23]. IR loss represents a key evolutionary adaptation in *Vicia*, contributing to both genome size reduction and structural rearrangements [22,24]. Notably, the cp genome of *V. sepium* illustrates this pattern, containing 110 genes and one pseudogene alongside specific gene and intron losses (*ycf4*, *clpP* intron, and *rpl16* intron deletions) and insertions (*rpl20* and *ORF292*) [24]. This trend extends to other *Vicia* species: *V. ramuliflora* (124,682 bp) maintains 109 genes [25], *V. kulingana* (125,696 bp) possesses 102 genes [26], and *V. cracca* (126,272 bp) retains 108 genes [27], all lacking IR regions. Indeed, despite the absence of IRs, core gene content remains remarkably conserved across these species, with protein-coding genes (75–77), tRNAs (28–30), and rRNAs (4) showing limited variability. In contrast, simple sequence repeat (SSR) abundance displays striking interspecific divergence; for instance, *V. bungei* contains 432 SSRs, over sixfold more than *V. sepium* (66 SSRs) [22,24], highlighting species-specific microsatellite accumulation patterns. These genomic modifications, particularly IR loss and SSR variability, likely influence evolutionary trajectories by altering gene dosage effects and mutation rates [28]. Collectively, these findings underscore the dynamic nature of cp genome evolution in *Vicia*, where structural simplification coexists with functional conservation and species-specific repeat diversification.

Despite significant progress in cp genome research, numerous economically and medicinally valuable *Vicia* species, particularly those indigenous to Iran, remain genomically uncharacterized. Notably, key species including *V. hirsuta, V. hybrida, V. lathyroides, V. lutea, V. narbonensis, V. peregrina*, and *V. villosa* currently lack complete cp genome sequences. This knowledge gap substantially hinders the comprehensive understanding of their genomic architecture, evolutionary relationships, and potential biotechnological utility. To address this critical research need, we present the first complete cp genome sequences for seven Iranian *Vicia* species, employing advanced sequencing and bioinformatic approaches. Our comprehensive investigation included: (1) high-quality genome assembly and annotation, (2) detailed structural characterization, and (3) comparative genomic analyses to identify both conserved and hypervariable regions with potential as molecular markers. Furthermore, we conducted systematic examinations of repetitive elements and codon usage patterns to elucidate evolutionary dynamics and functional adaptations. Phylogenetic reconstructions were performed to clarify taxonomic relationships within the genus and resolve evolutionary histories. This work establishes an essential genomic foundation for future studies on *Vicia* biodiversity, conservation genetics, and applied research. The newly generated cp genome data will facilitate: (1) precise species identification, (2) investigations of population genetics, and (3) exploration of adaptive mechanisms in this ecologically and economically significant legume genus. Moreover, these resources enable comparative genomic studies across Fabaceae and support efforts to harness *Vicia* genetic potential for agricultural improvement and medicinal applications.

## Materials and methods

### Plant material and DNA extraction

We collected leaf samples from seven *Vicia* species during their peak flowering period. The sampling encompassed distinct geographical locations across northern and central Iran. Specifically, four species, *V. hirsuta*, *V. hybrida*, *V. lathyroides*, and *V. lutea*, were sampled in the Golestan National Park near Gorgan. The remaining three species included *V. narbonensis* from Jajrood, Tehran Province, *V. peregrina* from Tapeh Abbas Abad, Hamedan Province, and *V. villosa*

from Varamin, Tehran Province. No special collection permits were required as these species are neither endangered nor protected in the sampling locations, and all collections were conducted in accordance with local regulations. Total genomic DNA was isolated from 100 mg of fresh leaf tissue using the DNeasy Plant Mini Kit (QIAGEN, Germany). The DNA purity and concentration were evaluated using 1% agarose gel electrophoresis and a NanoDrop 2000c spectrophotometer (Thermo Fisher Scientific, USA). Only high-quality DNA samples were selected for library preparation, which was carried out according to the manufacturer's protocol. The sequencing was conducted on an Illumina HiSeq 2000 platform (Illumina Inc., USA), generating paired-end reads of 150 bp in length.

## Cp genome assembly and annotation

The raw paired-end reads (150 bp in length) from each accession were initially evaluated for quality using FastQC v0.11.9 (available at: https://www.bioinformatics.babraham.ac.uk/projects/fastqc/). Adapter sequences and low-quality reads were removed using Trimmomatic v0.39 [29]. The processed high-quality reads were then assembled into complete cp genomes using GetOrganelle v1.7.7.1 [30]. To confirm genome integrity, the assembly graphs were examined and verified using Bandage [31]. For annotation, three tools were employed: CPGAVAS2 [32], GeSeq [33], and PGA [34]. Finally, circular visualizations of the cp genomes were generated with OGDRAW [35].

## Genome feature characterization

In this study, Simple Sequence Repeats (SSRs) were detected using the MISA Perl Script [36] with threshold parameters set to a minimum of eight repeats for mononucleotide SSRs, four repeats for di- and trinucleotide SSRs, and three repeats for tetra-, penta-, and hexanucleotide SSRs. Long repetitive sequences were further analyzed using REPuter [37](available at: http://bibiserv.techfak.uni-bielefeld.de/repeater/), with a minimum repeat size of 30 bp and a Hamming distance of three to ensure stringent identification of homologous regions. To assess codon usage bias, Relative Synonymous Codon Usage (RSCU) values were calculated using MEGA6 [38]. The RSCU patterns were visualized using an interactive RSCU plot generated with the RSCU-Plot Shiny app (available at: https://pcg-lab.shinyapps.io/RSCU-Plot/). Additionally, nucleotide diversity (Pi) for each gene was computed using CPStools [39], and the resulting data were visualized using R to facilitate comparative analysis of genetic variation.

To evaluate the evolutionary pressures acting on the chloroplast genomes, we conducted a selection pressure analysis across the shared protein-coding genes of the 16 *Vicia* species. This was performed using EasyCodeML [40], implementing a suite of site models to detect signatures of positive selection. We compared four nested model pairs (M0 vs. M3, M1a vs. M2a, M7 vs. M8, and M8a vs. M8) and employed likelihood ratio tests (LRTs) with a significance threshold of *p* < 0.05 to identify genes under diversifying selection. For each gene, the nonsynonymous-to-synonymous substitution rate ratio ($\omega$ = dN/dS) was calculated. Genes and specific codon sites were inferred to be under positive selection based on a combination of significantly elevated $\omega$ values and statistical support from the LRTs.

## Comparative cp genomic analysis

A robust comparative genomic analysis was conducted to elucidate cp genome structure across 16 *Vicia* species. The study encompassed seven newly sequenced *Vicia* species, supplemented with data from nine additional species retrieved from public databases: *V. bungei* (MT362055) [18], *V. costata* (NC_057995), *V. cracca* (MW266076) [27], *V. faba* (MT120813), *V. kulingana* (PQ576733) [26], *V. ramuliflora* (MN758738) [25], *V. sativa* (NC_027155), *V. sepium* (NC_039595) [24], and *V. tibetica* (OR491712). Cp genome organization was compared and visualized using the BLAST Ring Image Generator (BRIG) software [41]. Furthermore, the Mauve multiple genome alignment algorithm [42] was employed to identify structural variations, including rearrangements, and assess collinearity, among the *Vicia* cp genomes.

## Phylogenetic analysis

To elucidate the evolutionary relationships within the genus *Vicia*, we conducted a comprehensive phylogenetic analysis using coding sequences obtained from our sequenced species, supplemented with nine additional cp genomes retrieved from the NCBI database, collectively representing 16 *Vicia* species. *Cicer arietinum* was selected as the outgroup to root the phylogenetic tree. Initial sequence alignment was performed using MUSCLE v3.8.1551 [43] under default parameters to generate high-quality multiple sequence alignments. To enhance alignment accuracy, poorly aligned regions and gaps were removed using trimAl v1.4 [44] with stringent filtering criteria ("-gt 0.95 -st 0.001"). The refined alignments were subsequently concatenated into a single sequence matrix using SequenceMatrix [45], providing a unified dataset for phylogenetic reconstruction. Maximum likelihood analysis was executed in IQ-TREE [46] under the GTR + Gamma nucleotide substitution model, selected for its robustness in handling sequence evolution. Node support was evaluated through 1000 bootstrap replicates to assess the reliability of the inferred topology. The resulting phylogenetic tree was visualized and annotated using the Interactive Tree of Life (iTOL) platform [47], facilitating clear interpretation of evolutionary relationships among the studied taxa.

## Assessment of hypervariable marker efficacy

To assess the phylogenetic applicability of the proposed barcode regions *rps7* and *rpl20*, these loci were retrieved from a representative dataset spanning 16 *Vicia* species, with *C. arietinum* designated as the outgroup. Sequence alignments were generated, followed by stringent quality trimming and phylogenetic analysis using established bioinformatics workflows, as outlined in the "phylogenetic analysis" section.

## Results

### Cp genome assembly and annotation

High-quality sequencing reads were assembled *de novo* to generate complete, circularized cp genomes for all seven investigated *Vicia* species. The assembled genomes exhibited size variation ranging from 118,660 bp (*V. peregrina*) to 130,223 bp (*V. hybrida*), with intermediate sizes observed for *V. hirsuta* (124,717 bp), *V. lathyroides* (123,358 bp), *V. lutea* (123,404 bp), *V. narbonensis* (124,953 bp), and *V. villosa* (125,979 bp) (Fig 1). Structural annotation demonstrated distinct organizational patterns among the species. *Vicia villosa* maintained the ancestral angiosperm chloroplast structure, featuring a quadripartite organization comprising duplicate inverted repeat regions (IRa/IRb; 1,746 bp each) separating the large single-copy (LSC; 103,653 bp) and small single-copy (SSC; 18,831 bp) regions. In contrast, the remaining species (*V. hirsuta, V. hybrida, V. lathyroides, V. lutea, V. narbonensis*, and *V. peregrina*) exhibited the derived inverted-repeat-lacking clade (IRLC) architecture, characterized by complete loss of one IR copy. This structural reduction aligns with established evolutionary patterns observed in IRLC legumes, supporting the phylogenetic placement of these *Vicia* species within this clade.

The cp genomes of seven *Vicia* species were found to contain a conserved set of photosynthetic genes (Table S1), including five photosystem I subunits (*psaA, psaB, psaC, psaI, psaJ*), fourteen photosystem II subunits (*psbA-K*), six ATP synthase components (*atpA-F*), and eleven NADH dehydrogenase genes (*ndhA-K*). The cytochrome b/f complex was consistently represented by six genes (*petA-D, petG, petL, petN*), while the large subunit of Rubisco (*rbcL*) was present in all species. The self-replication machinery showed high conservation, with eight large ribosomal subunit proteins (*rpl2, rpl14, rpl16, rpl20, rpl23, rpl32, rpl33, rpl36*) and 11 small subunit proteins (*rps2–4, rps7–8, rps11–12, rps14–15, rps18–19*) identified across species. The DNA-dependent RNA polymerase subunits (*rpoA, rpoB, rpoC1, rpoC2*) were uniformly well-annotated. Transfer RNA complement ranged from 29–35 genes, with *V. villosa* showing the highest count (35) due to duplications in *trnM-CAU* (5 copies) and *trnN-GUU* (3 copies). The standard set of ribosomal RNAs (*rrn5S, rrn4.5S, rrn16S, rrn23S*) was complete in all species. *accD* was annotated in all species except *V. lathyroides*, where it appears to

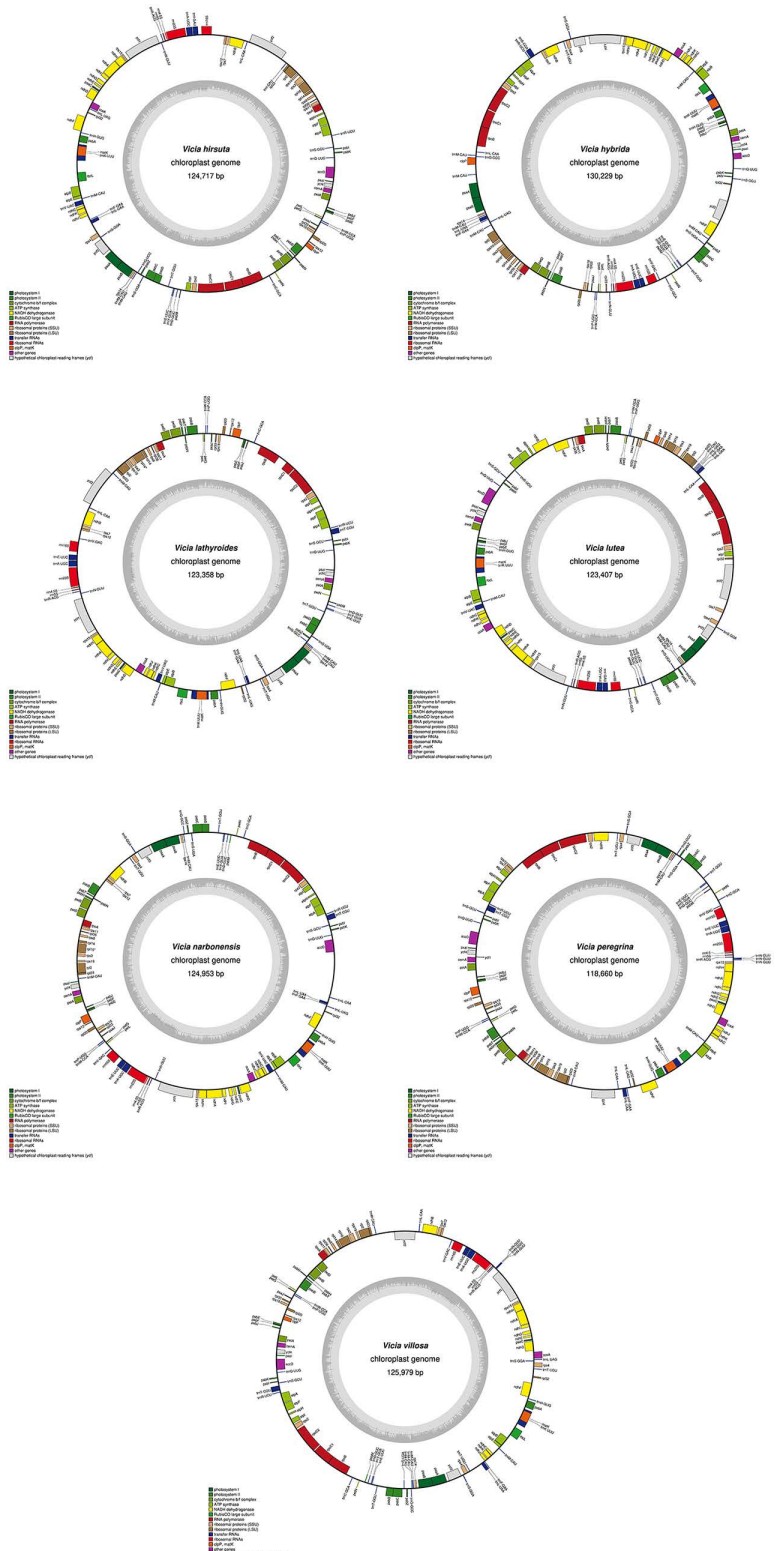

**Fig 1. Circular representations of *Vicia* cp genomes, generated using OGDRAW.** Key features, including gene locations and structural elements, are annotated to provide a comprehensive overview of the genome architecture.

be a lost gene. The hypothetical ORF *ycf2* was identified as a lost gene in *V. narbonensis*, while being properly annotated in other species. Other conserved functional genes included *ccsA*, *cemA*, *clpP* (with intron status varying by species), and *matK*, all of which were reliably annotated across the genus.

## Genomic feature comparison

Microsatellite characterization across seven *Vicia* species (Fig 2) demonstrated conserved genomic architecture with marked mononucleotide predominance (74–87% of total SSRs), particularly A/T-rich repeats, reflecting the genus' high AT-content. While *V. hybrida* contained the highest SSR density (77 total, including 16 compound microsatellites), all species shared similar repeat-type hierarchies (mono->di->tri-nucleotides) and dinucleotide bias toward AT/TA motifs (71–100% of dinucleotides). Notably, *V. hybrida* and *V. peregrina* exhibited elevated complex SSR counts (16 each), suggesting greater genomic plasticity, whereas *V. lutea* showed the simplest profile (45 mononucleotides, only 3 dinucleotides). The single hexanucleotide occurrence in *V. villosa* (CTCTTC) represents a rare departure from the predominant short-repeat architecture, potentially indicating species-specific transposable element activity. These patterns collectively highlight both deep conservation in microsatellite organization and subtle lineage-specific modifications in *Vicia* genome evolution.

Our analysis of repeat distributions across seven *Vicia* species revealed significant interspecific variation in repeat class composition and length distributions (Fig 3). The forward (F) class repeats dominated in most species, particularly in *V. hybrida* (n = 31 loci), *V. lutea* and *V. villosa* (n = 27), while *V. peregrina* showed the least number of F-class repeats (n = 19). *V. hybrida* exhibited the greatest length variation, with both extremely long (826 bp) and numerous short P-class repeats. Reverse (R) repeats were rare overall but showed species-specific patterns, with only two loci detected in *V. hirsuta* and complete absence in other species.

Analysis of RSCU in seven *Vicia* species revealed strong biases toward A/T-ending codons, consistent with the high AT content typical of plant genomes (Fig 4). Notably, leucine (UUA, RSCU ≈ 2.00) and isoleucine (AUU, RSCU ≈ 1.50–1.57) were universally preferred, while phenylalanine (UUU, RSCU ≈ 1.39–1.43) and tyrosine (UAU, RSCU ≈ 1.56–1.63) also showed high usage. Stop codon preferences varied slightly, with UAA dominant in most species (RSCU ≈ 1.12–1.35) but

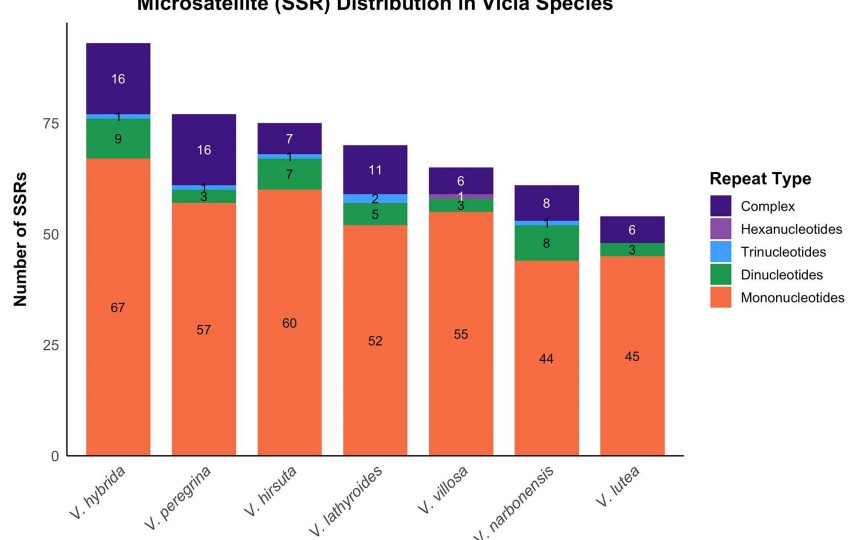

**Fig 2. The types and distribution of SSRs along the chloroplast genomes of seven *Vicia* species.**

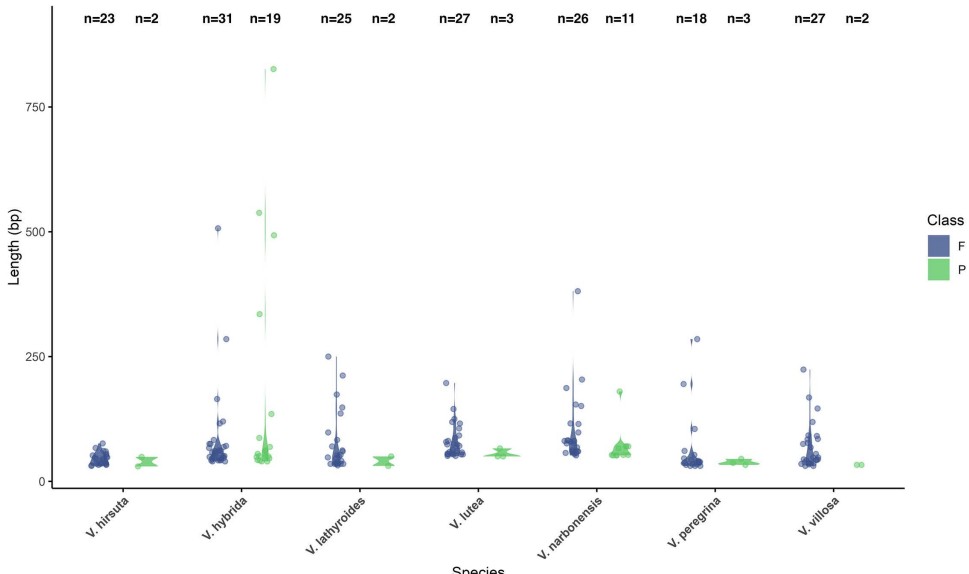

**Fig 3. Repeat length distributions by species and repeat class (Forward (F)=blue, Palindromic (P)=green) in *Vicia* taxa.** Points show individual loci; labels indicate counts (n=) per class-species group.

UGA elevated in *V. lathyroides* and *V. lutea* (RSCU ≈ 1.14–1.15). Arginine codons exhibited the strongest divergence, with AGA highly favored (RSCU ≈ 1.61–1.83) and CGG underrepresented (RSCU ≈ 0.52–0.64). Minor species-specific differences were observed, such as higher UCG (serine) usage in *V. villosa* (RSCU = 0.58) compared to *V. peregrina* (RSCU = 0.55), and slightly elevated GCG (alanine) in *V. hybrida* (RSCU = 0.51) versus *V. narbonensis* (RSCU = 0.47).

## Nucleotide diversity analysis

Analysis of cp genomes across seven *Vicia* species (Fig 5) revealed striking variation in nucleotide diversity (Pi), with the protease gene *clpP* (Pi = 0.19772) and hypothetical reading frame *ycf1* (Pi = 0.16964) showing the highest divergence. Ribosomal protein genes exhibited extreme variability (*rps7* [Pi = 0.09247], *rpl20* [Pi = 0.08962]), while core photosynthetic genes were highly conserved (*psbA* [Pi = 0.00646], *rbcL* [Pi = 0.01331]). Transfer RNAs displayed a broad range (*trnF-GAA* [Pi = 0.02712] vs *trnR-ACG* [Pi = 0.00386]), as did NADH dehydrogenase subunits (*ndhB* [Pi = 0.04259] vs *ndhE* [Pi = 0.00996]). RNA polymerase subunits (Pi = 0.02653–0.03861) and cytochrome b/f complex genes (Pi = 0.00501–0.03671) showed intermediate variation.

## Positive selection analysis of cp genes in *Vicia* species

Analysis of the chloroplast protein-coding genes across 16 *Vicia* species revealed significant signatures of positive selection in several genes, indicating a history of adaptive evolution (Table 1). The most pronounced signals were identified in genes encoding ribosomal proteins and components of photosynthetic and respiratory complexes, underscoring key functional categories that have undergone diversifying selection. Notably, multiple ribosomal protein genes exhibited strong evidence of positive selection, characterized by elevated nonsynonymous-to-synonymous substitution rate ratios (ω) and numerous codon sites with high posterior probabilities. These include *rps2*, *rps3*, *rps4*, *rps7*, and *rps18*, each harboring over ten positively selected sites. For instance, rps18 contained several sites under near-certain positive selection (e.g., 33Q, 37L, 87R, 121L), suggesting adaptive fine-tuning of the small ribosomal subunit. Similarly, the *cemA* gene, involved in carbon dioxide uptake and cytochrome c biogenesis, displayed one of the most extensive patterns of positive selection,

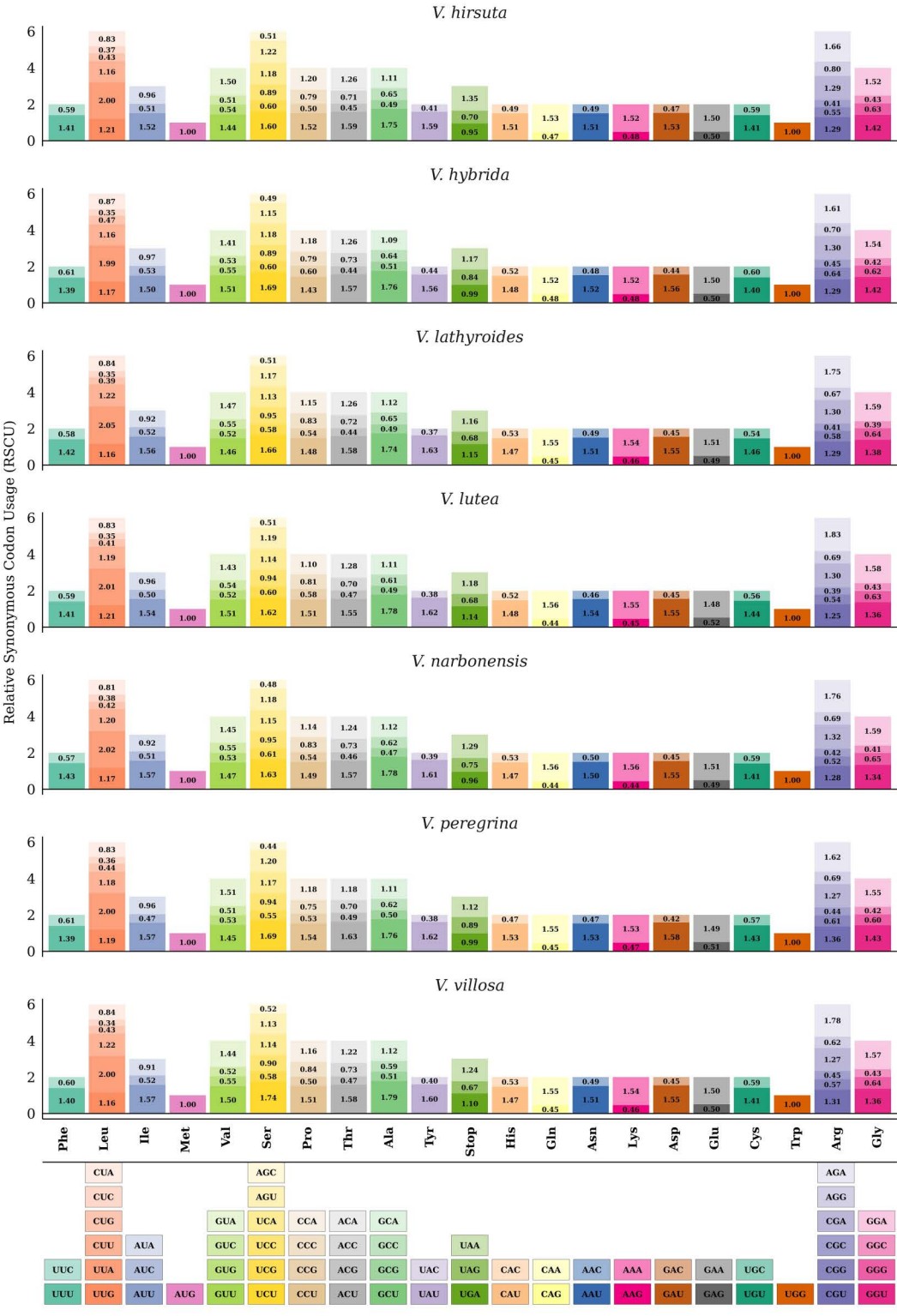

**Fig 4. The bar plot of the Relative Synonymous Codon Usage (RSCU) values for each amino acid, grouped by species.** Codons are color-coded, and their corresponding amino acids are labeled below the plot.

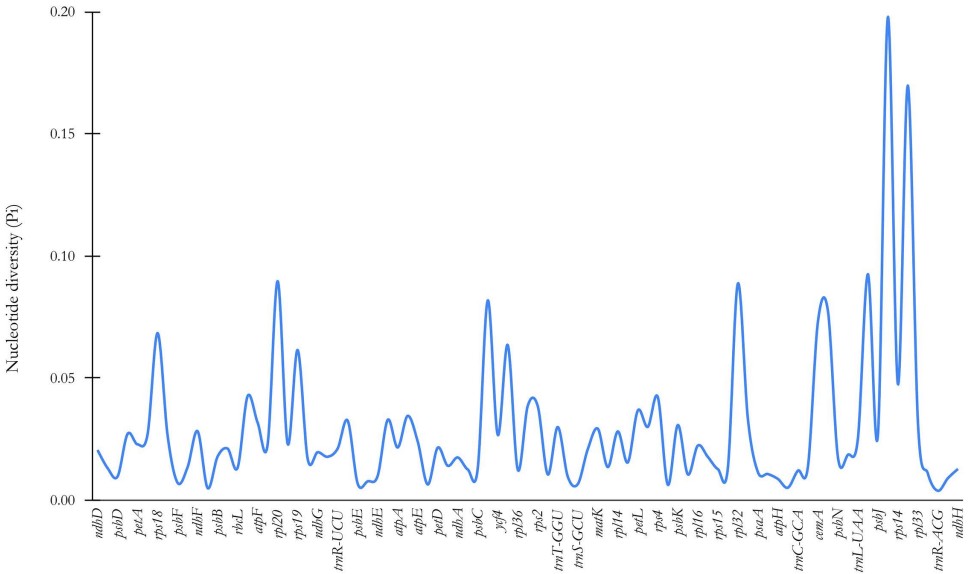

**Fig 5. Nucleotide diversity (Pi) across coding regions in the cp genomes of seven *Vicia* species.**

**Table 1. The results of positive selective pressure analysis in M7 vs. M8 model.**

| Gene | LnL | ω | LRT *p*-value | Positively Selected Sites |
|---|---|---|---|---|
| cemA | −2370.63 | 3.89 | < 0.001 | 2 A 0.896, 45 S 0.906, 55 F 0.989, *75 H 0.945, 98 C 0.943, 136 L 0.976*, 150 L 0.905, 156 I 0.921, 162 K 0.944, 181 Q 0.993, 185 V 0.995, 190 L 0.981, *196 T 0.975*, 197 L 0.961, *201 S 0.989*, 208 R 0.997, 215 V 1.000, 221 T 1.000** |
| rps18 | −1061.66 | 5.45 | < 0.001 | 6 S 0.986, *28 L 0.970*, 33 Q 1.000, 35 L 0.984*, 37 L 0.999, 77 S 1.000, 78 L 0.982, *80 A 0.976*, 85 E 0.996, 87 R 1.000, 98 N 0.996 |
| rps2 | −1965.27 | 4.41 | < 0.001 | 15 K 1.000, 20 F 0.999, 23 Y 0.975, *36 L 0.985*, 37 G 0.962*, 63 Y 0.996**, 79 K 0.982*, 83 S 0.969*, 116 R 0.992**, 117 Q 0.958*, 124 E 1.000, 127 T 0.976, *168 V 0.987*, 195 S 0.990, 215 R 0.980* |
| rps3 | −1648.06 | 4.88 | < 0.001 | 3 L 0.992, 5 N 0.999, 6 L 0.996, 9 N 0.999, 10 P 0.988, *11 E 0.996, 12 I 1.000, 17 H 0.997, 30 I 0.998, 31 N 0.952*, 33 A 0.990, *36 T 0.979*, 53 G 0.998, 60 P 0.979*, 134 I 0.999 |
| rps4 | −1631.34 | 6.86 | < 0.001 | 26 R 0.985, *31 G 1.000, 34 Q 0.995, 59 Q 0.976*, 68 A 0.967*, 146 S 0.996, 147 A 0.990, 150 K 0.999, 159 P 0.998, 160 S 1.000** |
| rps7 | −1796.03 | 6.98 | < 0.001 | 11 S 0.996, 48 Y 0.984, *56 Y 0.969*, 63 R 1.000, 64 E 1.000, 68 T 1.000, 70 A 0.995, 89 S 1.000, 110 Q 0.997, 121 L 1.000, 146 A 0.992, 147 F 0.982*, 150 I 0.999 |
| rps11 | −1467.02 | 5.55 | < 0.001 | 106 Y 1.000, 107 R 0.998 |
| rps19 | −822.52 | 5.94 | 0.0045 | 84 T 0.996** |
| ndhF | −5020.78 | 3.88 | < 0.001 | 46 L 0.964*, 652 M 0.992, 742 L 0.998, 744 L 0.990** |
| ndhH | −2167.36 | 4.97 | < 0.001 | 195 R 0.995** |
| petB | −1219.30 | 3.56 | < 0.001 | 1 S 0.997** |

*: p < 0.05,

**: p < 0.01

with 20 identified sites, including four with extremely high confidence (181Q, 185V, 215V, 221T). Beyond ribosomal and metabolic genes, significant positive selection was also detected in subunits of the NADH dehydrogenase complex. The *ndhF* gene contained four positively selected sites, with 652M and 742L showing particularly strong support. This pattern points toward potential adaptive evolution in the chloroplast's photosynthetic electron transport chain.

## Comparative analyses of cp genomes of *Vicia* species

A comprehensive comparison of sequence identity among 16 *Vicia* species identified distinct patterns of conservation and divergence, with protein-coding regions exhibiting significantly higher sequence identity (BLASTP) than noncoding regions (BLASTN), consistent with strong purifying selection on functional domains (Fig 6). Notably, several genes and intergenic regions displayed pronounced variability, including *rps11*, *rpoA*, *rps18*, *rpl32, rpl33, rpl23, ycf1, ycf2, rpoC1, clpP*, and *accD*, which contained segments with less than 70% sequence identity, suggesting potential hotspots for species-specific adaptation. Among noncoding regions, the spacers *rps15-ycf1*, *ycf1-trnN-GUU*, *rrn16-rps12*, *ycf2-trnI-CAU*, and *psbB-petL* were particularly divergent, likely due to reduced evolutionary constraints. Additionally, intronic regions such as the *rpl16* intron exhibited high variability, further highlighting the dynamic nature of noncoding sequences.

Despite the overall conservation of cp genomes, the comparative analysis of 16 *Vicia* species using Mauve (Fig S1) revealed multiple structural rearrangements, including inversions, translocations, and localized collinearity breaks. These variations are particularly pronounced in the Papilionoideae subfamily, to which *Vicia* belongs, and provide critical insights into the evolutionary history of this group. The most pronounced rearrangements were observed between distantly related species (*V. narbonensis* vs *V. villosa*), whereas closely related pairs (*V. sativa* vs *V. faba*) maintained near-identical synteny.

## Phylogenetic analysis of *Vicia* species

The seven *Vicia* species sequenced in this study (*V. hirsuta, V. hybrida, V. lathyroides, V. lutea, V. narbonensis, V. pere-grina,* and *V. villosa*) alongside additional *Vicia* species retrieved from public databases, and *Cicer arietinum* used as an

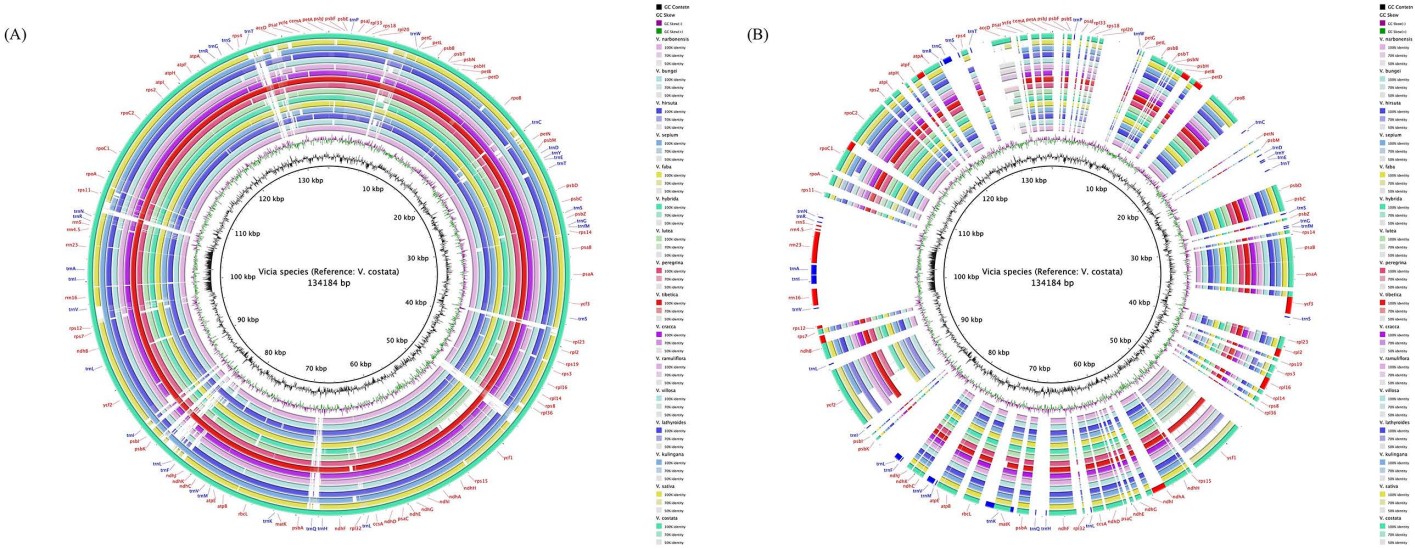

**Fig 6. Comparative sequence identity analysis among 16 *Vicia* species.** The reference genome is represented by the outer circle, while subsequent concentric circles illustrate pairwise sequence identity between *V. costata* and the 15 remaining species. **(A)** Nucleotide-level BLAST comparison. **(B)** Protein-level BLAST comparison.

outgroup, were analyzed based on 86 common genes to depict the evolutionary relationships (Fig 7). The results revealed that *V. peregrina, V. lutea, V. hybrida,* and *V. narbonensis* formed a strongly supported monophyletic clade (bootstrap support = 100), indicating that these species shared a recent common ancestor. This clade clustered closely with *V. faba*, a widely cultivated species, suggesting potential genetic similarities and shared evolutionary traits. Similarly, *V. hirsuta* and *V. villosa* formed another well-supported monophyletic group (bootstrap = 100), indicating a distinct lineage within *Vicia*. These two species clustered with *V. cracca* and *V. bungei*, suggesting that they belong to a broader evolutionary group with climbing or vining growth habits. In contrast, *V. lathyroides* was positioned separately from these clades but remained within the main *Vicia* lineage. Its placement, along with *V. sepium* and *V. sativa*, suggested a more divergent evolutionary history, possibly due to differences in ecological adaptation or genome evolution. Despite its separate clustering, the tree topology confirmed that *V. lathyroides* still belonged to the *Vicia* genus. The overall phylogenetic structure supported *Vicia* as a monophyletic genus, with high bootstrap values reinforcing the reliability of these relationships. However, the placement of certain species suggested possible paraphyly within specific subgroups, particularly in relation to species not included in this study. For example, *V. costata, V. tibetica,* and *V. ramuliflora* formed a distinct lineage, which diverged earlier from the other *Vicia* species, indicating a separate evolutionary trajectory. The consistently high bootstrap values (mostly 100) confirmed the robustness of these evolutionary inferences.

## Phylogenetic utility of *rps7* and *rpl20* in *Vicia* species

The phylogenetic trees reconstructed from the *rps7* and *rpl20* loci demonstrated strong congruence with the whole cp genome phylogeny (Fig 8), supporting their effectiveness as DNA barcodes for *Vicia* species. Both markers resolved the relationships among the seven studied species in a manner consistent with the reference tree. Notably, *V. narbonensis* and *V. peregrina* formed a well-supported clade across all analyses, while *V. villosa* and *V. cracca* consistently appeared as sister taxa and showed close relationship with *V. hirsuta*. The placement of *V. lathyroides* close to *V. sativa* and *V. sepium* further aligned with the whole-genome topology, reinforcing taxonomic relationships. Although minor variations in branch lengths were observed, the overall structure of the *rps7* and *rpl20* trees matched the whole-genome phylogeny, particularly in distinguishing major lineages. These findings confirm that both loci are reliable for species discrimination

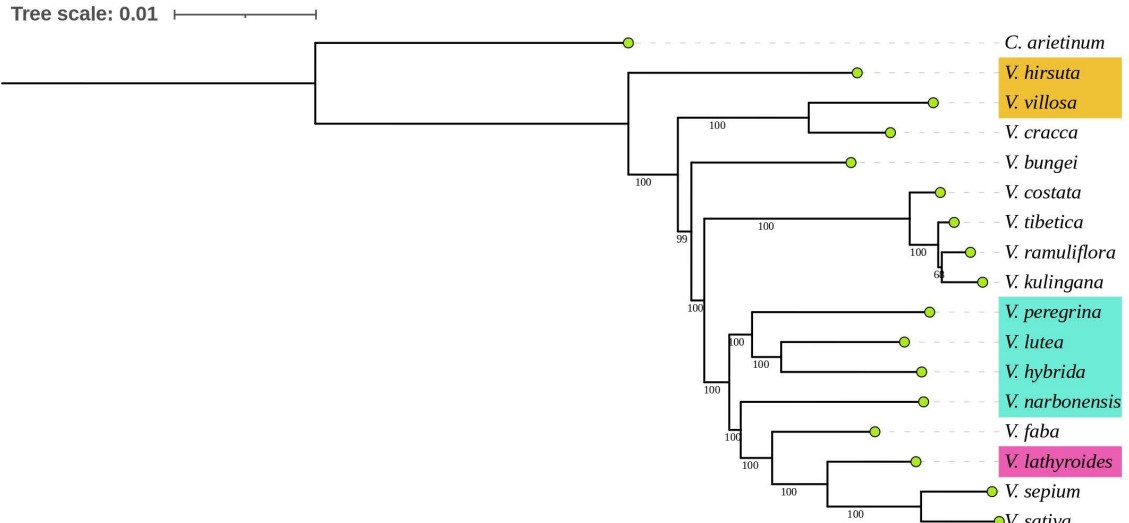

**Fig 7. Phylogenetic tree of 17 cp genomes, including 16 *Vicia* species and one outgroup (*C. arietinum*), reconstructed using maximum likelihood in IQ-TREE under the GTR + Gamma model.** The highlighted region emphasizes the position and relationships of the sample sequences from this study, illustrating their clustering with closely related species. Branch support was assessed with 100 bootstrap replicates.

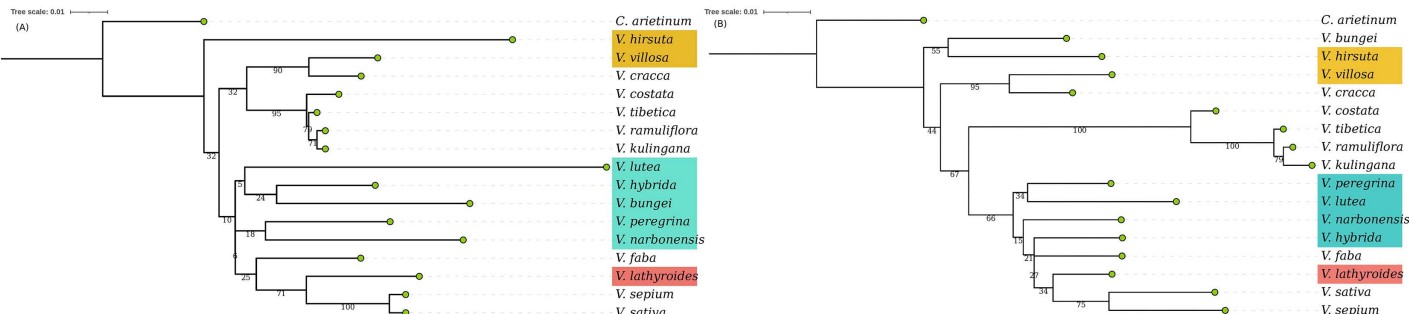

**Fig 8. Phylogenetic tree of 17 cp genomes, including 16 *Vicia* species and one outgroup (*C. arietinum*), reconstructed with (A) *rps7* and (B) *rpl20* using maximum likelihood in IQ-TREE under the GTR+Gamma model.** The highlighted region emphasizes the position and relationships of the sample sequences from this study, illustrating their clustering with closely related species. Branch support was assessed with 100 bootstrap replicates.

and phylogenetic reconstruction in *Vicia*, offering practical alternatives to full cp genome sequencing for future barcoding studies.

## Discussion

The structural divergence observed in *Vicia* cp genomes, notably the loss of one IR in six of the seven species, strongly supports their classification within the IR-lacking clade and highlights a key genomic synapomorphy for this group. This aligned with earlier studies on *V. sepium* (124,095 bp) and *V. bungei* (130,796 bp), which also exhibited IR loss [18,22]. The results indicated that the size differences are primarily driven by expansions or contractions in protein-coding regions, particularly in *accD*, *rps12*, and *ycf1*, as previously reported in Fabeae [48]. The loss of IR regions in *Vicia* species likely contributes to accelerated cp genome evolution through several mechanisms, including increased mutation rates, facilitated genome rearrangements, and altered selection pressures on intron structure [49]. This phenomenon is well-documented across legumes, with numerous studies reporting gene and intron losses similar to those observed in our study. For instance, chickpea (*Cicer arietinum*) shows loss of introns from *rps12* and *clpP* genes [50], *Trifolium* species have lost the *accD* gene [51,52], and *Vicia* species exhibit losses of *rpl22*, *rps16*, and one *clpP* intron [18]. These patterns of gene loss and structural simplification appear to be particularly prevalent in IRLC legumes, suggesting shared evolutionary trajectories following IR loss.

The lineage-specific losses of *accD* and *ycf2* further illustrate the genomic plasticity following IR loss, a phenomenon where non-essential genes are often pseudogenized or functionally relocated to the nucleus, as seen across the IRLC [53,54]. The variation in tRNA complement, including notable duplications in *V. villosa*, underscores the increased structural plasticity in IR-lacking genomes, a pattern also reflected in the unique ORF and pseudogene profiles of species like *V. sepium* [22]. These findings supported the hypothesis that IR loss led to increased structural plasticity, as seen in other IRLC taxa like *Lathyrus* and *Pisum* [50]. The loss of one *clpP* intron in the studied species (except *V. villosa*) further corroborated the IRLC's distinct evolutionary trajectory, as this reduction was widespread in the clade [55]. Similar intron losses in rpl16 (observed in *V. faba*) and *clpP* (in *Glycyrrhiza glabra*) suggested that these modifications might have arisen from relaxed selection pressures following functional gene transfers to the nucleus [56]. Additionally, the low GC content (34.7–35.1%) across *Vicia* cp genomes, including *V. sepium* and *V. ramuliflora*, may have reflected higher mutational rates in IR-lacking species [57].

The nucleotide diversity analysis revealed striking patterns of molecular evolution across the cp genomes of *Vicia* species. The protease gene *clpP* ($\pi = 0.19772$) and hypothetical reading frame *ycf1* ($\pi = 0.16964$) emerged as the most divergent loci, consistent with previous reports of their hypervariability in legumes [58]. These findings support their continued

use as primary DNA barcodes, while the intermediate variability observed in ribosomal protein genes *rps7* ($\pi$ = 0.09247) and *rpl20* ($\pi$ = 0.08962) identifies them as valuable supplementary markers. The phylogenetic trees reconstructed from these ribosomal protein genes showed strong congruence with the whole cp genome phylogeny, demonstrating their utility for species discrimination. The *clpP* gene, while useful for distinguishing some *Actinidiaceae* species [59], lacks universal resolution. Similarly, the *ycf1* gene, which showed the second-highest $\pi$ value in our analysis, has proven valuable for species identification in diverse plant groups [60] including *Pinus* [61] and *Orchidaceae* [62]. While the full-length *ycf1* (approximately 7000 bp) is too long for conventional barcoding applications [58], our results support the use of *rps7* and *rpl20* that have shown high discrimination power across land plants.

Our findings offer tangible genomic resources that can directly inform and enhance conservation strategies for *Vicia* biodiversity. The high-resolution phylogenetic framework and the validated DNA barcodes (*rps7* and *rpl20*) provide a reliable method for the precise identification of species and evolutionarily significant units (ESUs), which is a critical first step in prioritizing conservation targets [63]. The hypervariable regions identified, such as *clpP* and *ycf1*, along with the characterized SSR markers, are powerful tools for conducting population genetic studies. These markers can be used to assess genetic diversity, population structure, and gene flow across natural populations, identifying those that are genetically depauperate or isolated. Consequently, the complete chloroplast genomes presented here serve as a foundational reference for genotyping germplasm bank accessions, guiding the selection of material for seed banking (*ex situ* conservation) and enabling the management of wild populations (*in situ* conservation) to maximize the preservation of genetic diversity. By applying these genomic tools, conservation efforts can move beyond morphology-based identification and adopt a molecular-driven strategy to safeguard the adaptive potential and long-term survival of *Vicia* species.

Comprehensive phylogenetic analyses of *Vicia* species, integrating complete cp genome data and the well-supported barcoding loci *rps7* and *rpl20*, elucidated complex evolutionary relationships with significant taxonomic implications. Our whole-genome and multi-locus phylogenetic reconstructions consistently identified *V. sepium* and *V. sativa* as sister species, corroborating previous findings by Li et al. (2020) [24]. The sister relationship between *V. sepium* and *V. sativa* aligns with their shared morphological characteristics, including similar floral morphology and the presence of adaxially hairy styles, as previously documented by Schaefer et al. (2012) [64]. Notably, *V. faba* formed a stable clade with these species across all analyses, suggesting shared evolutionary pathways. Furthermore, the early-diverging lineage comprising *V. ramuliflora*, *V. tibetica*, and *V. costata* was robustly supported, reinforcing their status as phylogenetically distinct from core *Vicia* taxa. The basal position of *V. ramuliflora*, *V. tibetica*, and *V. costata* mirrors earlier proposals that these species represent an ancestral lineage with unique ecological adaptations [64]. The strong concordance between our chloroplast-based phylogenies and prior plastome studies [24,64] underscores the continued value of cp genome data for resolving phylogenetic relationships at shallow to moderate evolutionary depths within *Vicia*. However, the persistent challenges in determining species boundaries and resolving deeper nodes highlight the need for complementary nuclear genomic data to provide a more comprehensive understanding of the genus's evolutionary dynamics.

## Conclusion

This study successfully delineated the complete chloroplast genomes of seven previously uncharacterized Iranian *Vicia* species, achieving its primary aim of expanding genomic resources for this economically significant genus. Our analyses uncovered substantial structural divergence, primarily driven by the predominant loss of one inverted repeat region in six species—a derived feature aligning them with IRLC, while *V. villosa* retained the ancestral quadripartite structure. Through comparative genomics, we identified *clpP* and *ycf1* as hypervariable regions and established *rps7* and *rpl20* as highly effective barcoding markers, providing robust tools for species discrimination. Phylogenetic reconstruction based on 86 shared genes yielded a well-resolved topology, clarifying evolutionary relationships and affirming monophyletic groupings with high statistical support. The lineage-specific gene losses, including *accD* in *V. lathyroides* and *ycf2* in *V. narbonensis*, further underscore the dynamic nature of chloroplast evolution in Vicia. To translate these findings into applied outcomes,

we recommend: (1) adopting *rps7* and *rpl20* as standard markers for phylogenetic and barcoding studies; (2) integrating the characterized SSRs and variable loci into population genetic and diversity assessments; and (3) prioritizing functional investigation of divergent genes such as *clpP*, *ycf1*, *accD*, and *ycf2* to elucidate their roles in adaptive and agronomic traits. This study establishes a critical genomic foundation for future research in *Vicia* systematics, conservation, and molecular breeding. However, to move from sequence correlation to functional understanding, future studies should employ comparative transcriptomics to assess RNA editing patterns and proteomic analyses to verify the expression and functional status of the proteins encoded by these variable genes.

## Supporting information

**S1 Table. Genes predicted in the chloroplast genome of seven Vicia species#: Intron number, (n): Gene copy number.**
(PDF)

**S1 Fig. Gene map and MAUVE alignment of 16 *Vicia* chloroplast genomes.**
(PDF)

## Author contributions

**Conceptualization:** Aboozar Soorni.

**Formal analysis:** Mohammad Mehdi Golchini, Aboozar Soorni.

**Investigation:** Mohammad Mehdi Golchini, Aboozar Soorni.

**Methodology:** Mohammad Mehdi Golchini.

**Project administration:** Aboozar Soorni.

**Validation:** Aboozar Soorni.

**Writing – original draft:** Aboozar Soorni.

**Writing – review & editing:** Mohammad Mehdi Golchini.

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
