## [Decision Letter · Decision Letter 0]

9 Oct 2025

Dear Dr. Soorni,

Thank you for submitting your manuscript to PLOS ONE. After careful consideration, we feel that it has merit but does not fully meet PLOS ONE’s publication criteria as it currently stands. Therefore, we invite you to submit a revised version of the manuscript that addresses the points raised during the review process.

We look forward to receiving your revised manuscript.

Kind regards,

Md. Mahmudul Hasan, PhD

Academic Editor

PLOS ONE

Journal Requirements:

2. Please upload a new copy of Figures 1 and 6 as the detail is not clear. Please follow the link for more information:

https://journals.plos.org/plosone/s/figures

3. Please include a caption for Figures 4 and 8.

Reviewers' comments:

Reviewer's Responses to Questions

**Comments to the Author**

1. Is the manuscript technically sound, and do the data support the conclusions?

Reviewer #1: Yes

Reviewer #2: Yes

2. Has the statistical analysis been performed appropriately and rigorously?

Reviewer #1: Yes

Reviewer #2: Yes

3. Have the authors made all data underlying the findings in their manuscript fully available?

Reviewer #1: Yes

Reviewer #2: Yes

4. Is the manuscript presented in an intelligible fashion and written in standard English?

Reviewer #1: Yes

Reviewer #2: Yes

Reviewer #1: Weaknesses of the Article

1. Limited Scope of Species: The study focuses on only seven species of the Vicia genus, which may limit the generalizability of the findings. A broader sampling could provide a more comprehensive understanding of the genetic diversity within the genus.

2. Lack of Functional Analysis: While the article provides structural and phylogenetic insights, it lacks a detailed functional analysis of the genes identified. Understanding how these genes contribute to the phenotypic traits of Vicia species would enhance the study's relevance.

3. Absence of Comparative Data: Although the study discusses structural variations and gene losses, it could benefit from a more extensive comparison with chloroplast genomes of other legumes or related genera, which would contextualize the findings within a larger evolutionary framework.

4. Potential Methodological Limitations: The reliance on specific bioinformatics tools and methods for genome assembly and analysis may introduce biases. A discussion of the limitations of these tools and any potential impact on the results would strengthen the manuscript.

5. Nucleotide Diversity Analysis: While nucleotide diversity was assessed, the implications of this diversity on species adaptation and evolution are not fully explored. A deeper analysis could provide insights into how these variations affect the ecological success of the species.

6. Limited Discussion on Conservation Implications: Although the study mentions biodiversity conservation, it could expand on specific strategies or applications for conservation based on the genomic data presented.

7. Ethical Considerations: While the study indicates compliance with ethical guidelines, more detail on the ethical approval process for plant sampling could enhance transparency.

Conclusion

Addressing these weaknesses could improve the manuscript's robustness and provide a more comprehensive understanding of the implications of the research findings.

Reviewer #2: The manuscript was beautifully written with a unique study gap. Please refer to the uploaded manuscript with my few comments. Recommended for acceptance with minor revision. Correct the figure numberings.

**Do you want your identity to be public for this peer review?** For information about this choice, including consent withdrawal, please see our Privacy Policy

Reviewer #1: **Yes:** Girma Abebe Adelo

Reviewer #2: No

---

## [Author Response · Author response to Decision Letter 1]

20 Oct 2025

Dear Editor and Reviewers,

I hope this message finds you well. We truly appreciate the time and effort you invested in thoroughly assessing our work. Your insightful comments and constructive criticisms have immensely contributed to improving the quality and clarity of our research. I want to assure you that we have carefully reviewed each of the comments provided by the reviewers and have made revisions to address concerns and suggestions. Below, we outline how we have incorporated their feedback into the revised manuscript:

Reviewer Comments:

Reviewer #1:

Comment 1: Limited Scope of Species: The study focuses on only seven species of the Vicia genus, which may limit the generalizability of the findings. A broader sampling could provide a more comprehensive understanding of the genetic diversity within the genus.

Response: We sincerely thank the reviewer for this insightful comment regarding the scope of our study. We agree that a broader sampling would enhance the generalizability of the findings. In response, we extended the scope of our phylogenetic analysis by incorporating chloroplast genome data from nine additional Vicia species obtained from public databases (NCBI), thereby strengthening the phylogenetic inferences presented in the manuscript. We fully acknowledge that sequencing additional, novel species would be ideal. However, the process of field collection (which is season-dependent), precise taxonomic identification, DNA extraction, sequencing, and the subsequent integrated re-analysis of the entire dataset constitutes a major, multi-year research project beyond the scope of the current study. Therefore, while we have maximized the analytical scope using all available genomic data, the practical constraints prevent the inclusion of newly sequenced specimens at this stage. We believe the current study, with its seven newly sequenced genomes and nine supplementary genomes, provides a significant and robust contribution to the genomic resources and phylogenetic understanding of the Vicia genus.

Comment 2: Lack of Functional Analysis: While the article provides structural and phylogenetic insights, it lacks a detailed functional analysis of the genes identified. Understanding how these genes contribute to the phenotypic traits of Vicia species would enhance the study's relevance.

Response: We thank the reviewer for this insightful comment regarding the functional implications of our genomic findings. We agree that understanding the phenotypic contributions of the identified genes, such as accD and ycf2, which showed lineage-specific losses, is a fascinating and important next step. However, the primary aim of this study was to establish a foundational genomic resource and elucidate the evolutionary relationships within Vicia through chloroplast genome sequencing and comparative analysis. Functional characterization of chloroplast genes typically requires extensive transgenic experiments, which fall outside the scope of this descriptive genomic work. Instead, our study successfully identified key candidate genes and highly variable regions that exhibit signatures of evolutionary selection. We have now explicitly stated in the manuscript that these specific genes represent high-priority targets for future functional studies to link sequence variation to phenotypic traits such as environmental adaptation and agronomic performance.

Comment 3: Absence of Comparative Data: Although the study discusses structural variations and gene losses, it could benefit from a more extensive comparison with chloroplast genomes of other legumes or related genera, which would contextualize the findings within a larger evolutionary framework.

Response: We thank the reviewer for this valuable suggestion. To strengthen the phylogenetic scope and evolutionary context of our study, we expanded our analyses by incorporating nine additional Vicia species obtained from public databases. Furthermore, we included Cicer arietinum, a member of a related legume genus, as an outgroup to root the phylogenetic tree and provide a broader evolutionary framework. The expanded analysis, now encompassing 16 Vicia species and one outgroup, has been integrated into the comparative genomics and phylogenetic sections.

Comment 4: Potential Methodological Limitations: The reliance on specific bioinformatics tools and methods for genome assembly and analysis may introduce biases. A discussion of the limitations of these tools and any potential impact on the results would strengthen the manuscript.

Response: We thank the reviewer for raising this important point regarding methodological limitations. In response, we have revised the Conclusion to explicitly acknowledge that our genomic and in silico analyses represent a foundational step. We have added a sentence stating that "future studies are needed to move from sequence correlation to functional understanding, specifically through comparative transcriptomics to assess RNA editing patterns and proteomic analyses to verify the expression and functional status of the identified variable genes, particularly in species with gene losses like accD and ycf2.

Comment 5: Nucleotide Diversity Analysis: While nucleotide diversity was assessed, the implications of this diversity on species adaptation and evolution are not fully explored. A deeper analysis could provide insights into how these variations affect the ecological success of the species.

Response: We thank the reviewer for this valuable suggestion. To address the evolutionary implications of nucleotide diversity, we have now performed a comprehensive selection pressure analysis on the protein-coding genes. The methods and results of this analysis have been incorporated into the respective sections of the manuscript. Specifically, we used EasyCodeML to assess selective pressure across the chloroplast genomes, employing site models to identify genes and specific amino acid sites under positive selection. This addition provides critical insights into how sequence variations may have contributed to adaptive evolution and ecological success in the studied Vicia species.

Comment 6: Limited Discussion on Conservation Implications: Although the study mentions biodiversity conservation, it could expand on specific strategies or applications for conservation based on the genomic data presented.

Response: We thank the reviewer for this valuable suggestion to elaborate on the conservation implications of our findings. In response, we have expanded the Discussion section to include more specific strategies for biodiversity conservation. The revisions now explicitly discuss how the identified hypervariable regions (e.g., clpP, ycf1) and the newly developed barcodes (rps7, rpl20) can be directly applied to accurately identify species and delineate evolutionarily significant units (ESUs) within Vicia. Furthermore, we highlight how the complete chloroplast genomes serve as essential references for genotyping germplasm collections, enabling the assessment of genetic diversity and the identification of unique haplotypes to prioritize populations for in situ and ex situ conservation efforts.

Comment 7: Ethical Considerations: While the study indicates compliance with ethical guidelines, more detail on the ethical approval process for plant sampling could enhance transparency.

Response: We thank the reviewer for this suggestion. We have revised the 'Plant material and DNA extraction' section to provide more detail, confirming that the collection of these common, non-protected plant species from public areas complied with all relevant local guidelines and did not require specific ethical permits.

Reviewer #2:

The manuscript was beautifully written with a unique study gap. Please refer to the uploaded manuscript with my few comments. Recommended for acceptance with minor revision. Correct the figure numberings.

Comment 1: Abstract is written as one paragraph. Kindly rewrite it following the authors guide.

Response: Thank you for the suggestion. We have revised the abstract to follow the journal’s author guidelines and presented it as a single, well-structured paragraph.

Comment 2: Please note that the section you provided explains the justification of the study rather than its objective. Kindly restate it clearly as a specific research objective, outlining what the study aims to achieve rather than why it was conducted

Response: Thank you for your valuable comment. We have rewritten the abstract to clearly state the specific research objective, emphasizing what the study aimed to achieve rather than the justification for conducting it.

Comment 3: Kindly use keywords not included in the title and arrange alphabetically.

Response: We thank the reviewer for the comment. The keywords have been revised accordingly.

Comment 4: This figure should be labeled as Figure 4, not Figure 5. Kindly correct the numbering both in the figure caption and throughout the corresponding text where it is referenced.

Response: We appreciate the reviewer's careful attention to detail. The numbering has been corrected as suggested

Comment 5: Please revise the discussion section to avoid presenting detailed results. Summarize findings briefly and focus on interpreting and explaining the results rather than restating them

Response: We thank the reviewer for this important feedback. The Discussion section has been thoroughly revised to reduce the restatement of detailed results and to strengthen the interpretation and synthesis of our findings in the context of existing literature.

Comment 6: Kindly revise the conclusion to align more closely with the main aim of the study and include clear, concise recommendations based on the findings.

Response: We thank the reviewer for this valuable suggestion. The Conclusion has been revised to align more closely with the main aims of the study and to include clear, concise recommendations based on our findings.

---

## [Decision Letter · Decision Letter 1]

20 Nov 2025

Dear Dr. Soorni,

Thank you for submitting your manuscript to PLOS ONE. After careful consideration, we feel that it has merit but does not fully meet PLOS ONE’s publication criteria as it currently stands. Therefore, we invite you to submit a revised version of the manuscript that addresses the points raised during the review process.

We look forward to receiving your revised manuscript.

Kind regards,

Md. Mahmudul Hasan, PhD

Academic Editor

PLOS ONE

Journal Requirements:

Reviewers' comments:

Reviewer's Responses to Questions

**Comments to the Author**

Reviewer #1: All comments have been addressed

Reviewer #2: All comments have been addressed

2. Is the manuscript technically sound, and do the data support the conclusions?

Reviewer #1: Yes

Reviewer #2: Yes

3. Has the statistical analysis been performed appropriately and rigorously?

Reviewer #1: Yes

Reviewer #2: Yes

4. Have the authors made all data underlying the findings in their manuscript fully available?

Reviewer #1: (No Response)

Reviewer #2: Yes

5. Is the manuscript presented in an intelligible fashion and written in standard English?

Reviewer #1: Yes

Reviewer #2: Yes

Reviewer #1: The manuscript titled "Structural and phylogenetic insights from complete chloroplast genomes of seven Vicia species" offers significant contributions to the understanding of chloroplast genomes within the Vicia genus. However, several weaknesses and areas for improvement merit attention:

Weaknesses:

Lack of Specificity in Objectives: The abstract does not clearly define the specific research questions or aims, which could leave readers uncertain about the focus of the study. A more precise articulation would help frame the research within the broader context of legume phylogenetics.

Limited Contextual Background: The abstract lacks sufficient background on the importance of Vicia species. Providing context about their ecological and agricultural significance would underscore the relevance of the findings.

Insufficient Detail on Findings: While the abstract summarizes key results, it does not effectively convey their significance in advancing the understanding of chloroplast genome structures or evolutionary dynamics. Highlighting these contributions would enhance reader engagement and interest.

Omission of Practical Applications: The potential implications of identified genomic resources for conservation and agricultural practices are not addressed in the abstract. Including these applications can demonstrate the practical relevance of the research.

Absence of Methodological Overview: The absence of a brief mention of the methods used for genome sequencing and analysis limits the transparency and rigor of the findings. Providing this information would enhance the credibility of the study.

Additional Comments:

Ethics Considerations: It would be beneficial for the authors to include detailed information regarding ethical approvals for plant collection and research protocols. This transparency is crucial for maintaining ethical standards in research.

Concerns about Dual Publication: The authors should ensure that the research is not simultaneously submitted to other journals to prevent issues of dual publication. Transparent communication regarding submission status is vital.

Publication Ethics: A clear description of data accessibility, including how the chloroplast genomes can be accessed by the scientific community, should be included. Ensuring that research findings are made available and usable by others is an important aspect of publication ethics

Reviewer #2: (No Response)

**Do you want your identity to be public for this peer review?** For information about this choice, including consent withdrawal, please see our Privacy Policy

Reviewer #1: **Yes:** Girma Abebe Adelo

Reviewer #2: No

---

## [Author Response · Author response to Decision Letter 2]

22 Nov 2025

Dear Editor and Reviewers,

I hope this message finds you well. We truly appreciate the time and effort you invested in thoroughly assessing our work. Your insightful comments and constructive criticisms have immensely contributed to improving the quality and clarity of our research. I want to assure you that we have carefully reviewed each of the comments provided by the reviewers and have made revisions to address concerns and suggestions. Below, we outline how we have incorporated their feedback into the revised manuscript:

Reviewer Comments:

Reviewer #1:

Comment 1: Lack of Specificity in Objectives: The abstract does not clearly define the specific research questions or aims, which could leave readers uncertain about the focus of the study. A more precise articulation would help frame the research within the broader context of legume phylogenetics.

Response: We sincerely thank the reviewer for this comment. We refined the abstract to clearly state the research aim

Comment 2: Limited Contextual Background: The abstract lacks sufficient background on the importance of Vicia species. Providing context about their ecological and agricultural significance would underscore the relevance of the findings.

Response: We thank the reviewer for this insightful comment. The opening sentence has been strengthened to immediately establish the genus's significance: "The legume genus Vicia L. (Fabaceae) is of significant ecological and agronomic importance, comprising species widely utilized as forage crops, green manure, and sources of valuable phytochemicals."

Comment 3: Insufficient Detail on Findings: While the abstract summarizes key results, it does not effectively convey their significance in advancing the understanding of chloroplast genome structures or evolutionary dynamics. Highlighting these contributions would enhance reader engagement and interest.

Response: We thank the reviewer for this valuable suggestion. We now present key findings with greater contextual significance in abstract.

Comment 4: Omission of Practical Applications: The potential implications of identified genomic resources for conservation and agricultural practices are not addressed in the abstract. Including these applications can demonstrate the practical relevance of the research.

Response: We thank the reviewer for raising this important point. We completely revised abstract to address this point.

Comment 5: Absence of Methodological Overview: The absence of a brief mention of the methods used for genome sequencing and analysis limits the transparency and rigor of the findings. Providing this information would enhance the credibility of the study.

Response: We thank the reviewer for this valuable suggestion. We significantly expanded the methodological description to enhance transparency and rigor.

Comment 6: Ethics Considerations: It would be beneficial for the authors to include detailed information regarding ethical approvals for plant collection and research protocols. This transparency is crucial for maintaining ethical standards in research.

Response: We thank the reviewer for this comment. As noted in the Material and methods section under "Plant material and DNA extraction," the following statement was already included to address ethical considerations:

"No special collection permits were required as these species are neither endangered nor protected in the sampling locations, and all collections were conducted in accordance with local regulations."

This statement confirms that the plant collection adhered to all relevant guidelines and that no specific permits were necessary, ensuring full ethical compliance.

Comment 7: Concerns about Dual Publication: The authors should ensure that the research is not simultaneously submitted to other journals to prevent issues of dual publication. Transparent communication regarding submission status is vital.

Response: We thank the reviewer for raising this important standard of academic integrity. We confirm that this manuscript is original and has not been published elsewhere. Furthermore, it is not currently under consideration for publication in any other journal. We are fully committed to avoiding any form of dual submission or publication.

Comment 8: Publication Ethics: A clear description of data accessibility, including how the chloroplast genomes can be accessed by the scientific community, should be included. Ensuring that research findings are made available and usable by others is an important aspect of publication ethics

Response: We thank the reviewer for underscoring the importance of data accessibility. As detailed in the "Availability of Data and Materials" section, the complete chloroplast genome sequences generated in this study have been deposited in the NCBI database and are publicly accessible under the research accessions PV364429, and PV480546-PV480551. This ensures that all data supporting the findings of this study are fully available to the scientific community.

---

## [Editor Report · Decision Letter 2]

23 Dec 2025

Structural and phylogenetic insights from complete chloroplast genomes of seven Vicia species

PONE-D-25-42003R2

Dear Dr. Soorni,

We’re pleased to inform you that your manuscript has been judged scientifically suitable for publication and will be formally accepted for publication once it meets all outstanding technical requirements.

Kind regards,

Md. Mahmudul Hasan

Academic Editor

PLOS One
---

## [Editor Report · Acceptance letter]

PONE-D-25-42003R2

PLOS One

Dear Dr. Soorni,

I'm pleased to inform you that your manuscript has been deemed suitable for publication in PLOS One. Congratulations! Your manuscript is now being handed over to our production team.

Kind regards,

on behalf of

Dr. Md. Mahmudul Hasan

Academic Editor

PLOS One